# 4-Pyridone-3-carboxamide-1-β-D-ribonucleoside Reduces Cyclophosphamide Effects and Induces Endothelial Inflammation in Murine Breast Cancer Model

**DOI:** 10.3390/ijms26010035

**Published:** 2024-12-24

**Authors:** Paulina Mierzejewska, Agnieszka Denslow, Diana Papiernik, Alicja Zabrocka, Barbara Kutryb-Zając, Karol Charkiewicz, Alicja Braczko, Ryszard T. Smoleński, Joanna Wietrzyk, Ewa M. Słomińska

**Affiliations:** 1Department of Biochemistry, Medical University of Gdansk, 80-211 Gdańsk, Poland; b.kutryb-zajac@gumed.edu.pl (B.K.-Z.); alicja.braczko@gumed.edu.pl (A.B.); rt.smolenski@gumed.edu.pl (R.T.S.); 2Laboratory of Experimental Anticancer Therapy, Ludwik Hirszfeld Institute of Immunology and Experimental Therapy, Polish Academy of Sciences, 02-103 Wrocław, Polanddiana.papiernik@hirszfeld.pl (D.P.); joanna.wietrzyk@hirszfeld.pl (J.W.); 3Regional Center for Blood Donation and Blood Treatment in Gdansk, 80-309 Gdańsk, Poland; azzabrocka@gmail.com; 4Department of Perinatology and Obstetrics, Medical University of Bialystok, 15-089 Białystok, Poland; karol.charkiewicz@umb.edu.pl

**Keywords:** breast cancer, nicotinamide metabolism, cyclophosphamide

## Abstract

4-pyridone-3-carboxamide-1-β-D-ribonucleoside (4PYR) is a nicotinamide derivative, considered a new oncometabolite. 4PYR formation induced a cytotoxic effect on the endothelium. Elevated blood 4PYR concentration was observed in patients with cancer. Still, little is known about the metabolic and functional effects of 4PYR in this pathology. The study aimed to investigate whether this toxic accumulation of 4PYR may affect the activity of anticancer therapy with cyclophosphamide in the orthotropic model of breast cancer. Female Balb/c mice were injected with 4T1 breast cancer cells and assigned into three groups: treated with PBS (Control), cyclophosphamide-treated (+CP), 4PYR-treated (+4PYR), and mice treated with both 4PYR and CP(+4PYR+CP) for 28 days. Afterward, blood and serum samples, liver, muscle, spleen, heart, lungs, aortas, and tumor tissue were collected for analysis of concentrations of nucleotides, nicotinamide metabolites, and 4PYR with its metabolites, as well as the liver level of cytochrome P450 enzymes. 4PYR treatment caused elevation of blood 4PYR, its monophosphate and a nicotinamide adenine dinucleotide (NAD+) analog—4PYRAD. Blood 4PYRAD concentration in the +4PYR+CP was reduced in comparison to +4PYR. Tumor growth and final tumor mass were significantly decreased in +CP and did not differ in +4PYR in comparison to Control. However, we observed a substantial increase in these parameters in +4PYR+CP as compared to +CP. The extracellular adenosine deamination rate was measured to assess vascular inflammation, and it was higher in +4PYR than the Control. Treatment with 4PYR and CP caused the highest vascular ATP hydrolysis and adenosine deamination rate. 4PYR administration caused significant elevation of CYP2C9 and reduction in CYP3A4 liver concentrations in both +4PYR and +4PYR+CP as compared to Control and +CP. In additional experiments, we compared healthy mice without cancer, treated with 4PYR (4PYR w/o cancer) and PBS (Control w/o cancer), where 4PYR treatment caused an increase in the serum proinflammatory cytokine expression as compared to Control w/o cancer. 4PYR accumulation in the blood interferes with cyclophosphamide anticancer activity and induces a pro-inflammatory shift of endothelial extracellular enzymes, probably by affecting its metabolism by cytochrome P450 enzymes. This observation may have crucial implications for the activity of various anticancer drugs metabolized by cytochrome P450.

## 1. Introduction

Breast cancer is the most frequently diagnosed form of cancer in women. Recently developed targeted therapies, together with early detection, have considerably reduced breast cancer mortality in the last two decades. However, breast cancer is still one of the major leading causes of cancer-related death in women [1]. Triple-negative breast cancer (TNBC), which is clinically defined by the absence of expression of human epidermal growth factor receptor 2 (HER2), progesterone receptor (PR), and estrogen receptor (ER), contributes to approximately 15–20% of all breast cancer cases. This is one of the most aggressive breast cancer subtypes that is linked with poor prognosis. Therefore, new therapeutic approaches and a better understanding of the mechanisms of breast cancer progression are needed [2].

One of the most successful anticancer agents widely used as a chemotherapeutic agent is cyclophosphamide (CP), a nitrogen mustard derivative. It is used as a cytostatic drug from the group of alkylating drugs. The biological action is based on the interaction of the alkylating metabolites of CP with DNA, which leads to its fragmentation and, consequently, to cell death [3]. The chemical design of cyclophosphamide—substitution of an oxazaphosphorine ring for the methyl group of nitrogen mustard—was based on the fact that some cancer cells express high levels of phosphamidase, which is capable of cleaving the phosphorus–nitrogen (P—N) bond, releasing nitrogen mustard. Therefore, CP was one of the first drugs designed to target cancer cells selectively [4]. As an inactive prodrug, CP must be bioactivated in the liver into an active component and an inactive but toxic byproduct, acrolein, to provide its therapeutic effects. The CP action mechanism begins with the metabolic conversion of the prodrug into 4-hydroxycyclophosphamide, which exists in a state of equilibrium with its isomer, aldophosphamide [5]. Hepatic enzymes—the cytochrome P450 superfamily—are responsible for the formation of 4-hydroxycyclophosphamide (Figure 1) [6]. Cyclophosphamide, in combination with other antineoplastic agents, is used for the treatment of various cancers, including breast malignancies [7]. Even though CP is a commonly used anticancer and immunosuppressive drug, it unfortunately causes severe side effects, including cardiotoxicity. It has been suggested that phosphoramide is the active tumorigenic factor, and acrolein is a toxic metabolite that acts on endothelial cells and myocardium [8]. CP metabolites are linked with endothelial dysfunction, cardiomyocyte apoptosis, inflammation, calcium dysregulation, mitochondrial dysfunction, and endoplasmic reticulum damage [9,10].

Our earlier studies identified a new family of metabolites, derivatives of 4-pyridone-3-carboxamide-1B-D-ribonucleoside (4PYR). 4PYR, a nicotinamide metabolite, was first mentioned in 1979 in the urine of a patient with chronic megaloblastic leukemia [11]. Moreover, this molecule has been identified to accumulate in chronic renal disease [12], in active HIV infection [13], as well as non-small cell lung cancer [14] and renal cell carcinoma [15]. Our group discovered that 4PYR could be converted to triphosphates (4PYTP), diphosphates (4PYDP), monophosphates (4PYMP) but also to an analog of nicotinamide adenine dinucleotide (NAD)—4PYRAD. Formation of 4PYR nucleotides drains the energy and could be toxic, resulting in decreased ATP concentration and other metabolic abnormalities. Our previous studies on mice with 4T1 breast cancer demonstrated an early rise in blood 4PYR concentration soon after cancer cell injection. Moreover, we showed that 4PYR toxicity concerns, to a large extent, the vascular endothelium, the proper functioning of which is crucial for the prevention of cancer progression and metastasis [16] Given these observations and the fact that cyclophosphamide, in combination with other antineoplastic agents, is used to treat various cancers, including breast malignancies, in the current work, we aimed to investigate whether this toxic accumulation of 4PYR may affect the activity of anticancer therapy with cyclophosphamide in the orthotropic model of breast cancer.

## 2. Results

In this work, to investigate the effect of 4PYR on anticancer therapy with cyclophosphamide, we used only tumor-bearing animals (control), treated with cyclophosphamide (+CP), 4PYR (+4PYR), or a combination of these compounds (+4PYR+CP).

The concentration of 4PYR and its metabolites has been determined in the blood and the tissues: liver, muscles, spleen, heart, lungs, and brain. Most importantly, we observed an accumulation of blood 4PYR in Control. Cyclophosphamide anticancer treatment significantly reduced the concentration of blood 4PYR compared to Control +4PYR and +4PYR+CP groups, which were characterized by considerably higher blood 4PYR levels compared to Control and +CP. 4PYR derivatives (4PYMP and 4PYRAD) were accumulated in the 4PYR-treated groups: +4PYR and +4PYR+CP. There was no presence of 4PYR metabolites in the Control and +CP groups (Figure 2a).

Liver, lungs, and spleen were characterized by a more significant accumulation of 4PYMP than 4PYRAD in both 4PYR-treated groups: +4PYR and +4PYR+CP. The muscles of both +4PYR and +4PYR+CP showed a high concentration of 4PYMP but no 4PYRAD. On the other hand, accumulation of 4PYRAD but no 4PYMP was observed in the brain. All the examined tissues, except the lungs, were characterized by a lower concentration of both 4PYR derivatives, both 4PYMP and 4PYRAD, in the +4PYR+CP group compared to +4PYR (Figure 2b,c).

Tumor volume was measured every 48 h starting the 4th day after cancer cell implantation. Cyclophosphamide treatment caused significantly decreased tumor growth (Figure 3a) and final tumor mass (Figure 3b). Treatment with 4PYR did not affect tumor growth or mass compared to Control. The most interesting is that treatment with CP and 4PYR abolished the antitumor activity of CP (Figure 3a,b). However, we observed a significant increase in both of these parameters in +4PYR+CP as compared to +CP, suggesting that 4PYR exposure abolished the antitumor activity of CP.

To further characterize the effect of 4PYR accumulation on cancer and CP antitumor action, the ATP hydrolysis (ecto-nucleoside triphophate diphosphohydrolase (eNTPD)), AMP hydrolysis (ecto-5′-nucleotidase (e5′NT)), as well as adenosine deamination rate (ecto-adenosine deaminase (eADA)) on the aortas of Control, +CP, +4PYR and +4PYR+CP were measured. CP treatment did not affect vascular ATP, AMP hydrolysis, or adenosine deamination rate. We observed a reduced vascular ATP hydrolysis rate, a tendency to decrease the AMP hydrolysis rate but a significantly increased adenosine deamination rate on the aortic surface of the +4PYR group compared to the Control. There were no differences in vascular AMP hydrolysis rate in +4PYR+CP compared to all study groups. Treatment with +4PYR+CP was characterized by the highest vascular ATP hydrolysis and, in particular, vascular adenosine deamination rate, suggesting vascular inflammation (Figure 4).

In the next step, changes in the nicotinamide metabolism, reflecting the effect of 4PYR and CP treatment, were determined. CP treatment caused increased nicotinamide (NA) concentration and a tendency towards higher N-methylnicotinamide (MetNA) levels but decreased nicotinamide riboside (NR) levels in mice serum. We also observed reduced formation of final products of nicotinamide metabolism—N-methyl-2-pyridone-5-carboxamide (Met2PY) and N-methyl-4-pyridone-3-carboxamide (Met4PY) in +CP group. There was a tendency towards lower serum 4PYR in +CP group in comparison to control (Figure 5).

Moreover, 4PYR treatment contributed to an increase in the concentration of NA, a tendency towards a higher concentration of MetNA, a significantly lower NR level, as well as a considerably increased accumulation of one of the final products of nicotinamide metabolism—Met2PY, with a noticeable reduction in the concentration of the other—Met4PY, as compared to Control. Most importantly, the combined treatment of 4PYR with CP reduced the concentration of NA and MetNA compared to both +CP and +4PYR groups. We also observed a considerable decrease in NR concentration in the serum of +4PYR+CP mice compared to control, +CP, and +4PYR. +4PYR+CP mice were characterized by significantly enhanced concentration of Met2PY and Met4PY as compared to +CP and Control, but reduced concentration as compared to 4PYR (Figure 5).

Cyclophosphamide is metabolized in the liver to active metabolites by cytochrome P450 enzymes. To investigate whether the effect of 4PYR on the activity of this anticancer drug could be related to the disruption of hepatic CP metabolism by this nicotinamide metabolite, we examined the concentration of some of the cytochrome P450 enzymes: CYP2A6, CYP2C9, and CYP3A4 in the livers of mice treated with CP, 4PYR, or both CP and 4PYR (Figure 6). 4PYR treatment affected the liver concentration of cytochrome P450 enzymes involved in cyclophosphamide metabolism, in particular CYP2C9 (Figure 6b), the concentration of which was significantly higher in both the +4PYR and +4PYR+CP groups, and CYP3A4 (Figure 6c), the concentration of which was reduced in +4PYR and +4PYR+CP compared to the Control and +CP.

In additional experiments, we examined the expression of cytokines related to inflammation and tumor progression [17,18,19] in the serum of healthy mice without cancer treated with 4PYR. 4PYR treatment caused a significant increase in the expression of cytokines related to the inflammatory response: IL-1α, IL-1β, CCL19, CX3CL1, CCL3, also referred to as macrophage inflammatory protein 1α and CCL5, as well as FAS-ligand and SDF-1α in control mice without cancer as compared to the untreated group (Figure 7).

## 3. Discussion

This study revealed for the first time that accumulation of 4PYR and its intracellular metabolites may interfere with the action of anticancer treatment using cyclophosphamide as an example. We observed a reduced accumulation of 4PYR and its metabolites in the blood and organs of CP-treated mice. On the other hand, prolonged 4PYR exposure abolished the anticancer activity of cyclophosphamide and caused increased tumor volume and final tumor mass in the group treated with both 4PYR and cyclophosphamide. We also noted significantly increased extracellular nucleotide metabolism, particularly vascular ecto-adenosine deaminase activity on the vessel surface of 4PYR-treated mice receiving cyclophosphamide. Moreover, exogenous 4PYR administration affects the concentration of cytochrome P450 proteins involved in cyclophosphamide metabolism.

Our previous studies have shown the accumulation of 4PYR in mice blood as early as two days after tumor cell injection. Moreover, we demonstrated that the increase in blood concentration of 4PYR and the accumulation of its intracellular metabolites in cancer may facilitate disease progression. Our earlier studies highlighted that the increase in blood concentration of 4PYR and the accumulation of its intracellular metabolites in cancer may facilitate disease progression. Higher 4PYR level was linked to enhanced lung metastasis with mechanistic involvement of endothelial dysfunction in a murine intravenous breast cancer model [16]. We observed increased permeability of endothelial cells, reduced vascular relaxation ability after 4PYR prolonged exposure, and the impairment of L-arginine metabolism, as well as a higher ratio of vascular adenosine deamination rate, which is related to vascular inflammation [20].

This study was the next step in investigating the 4PYR metabolic and functional effects in cancer. Having proven that 4PYR accumulates in the course of cancer in mice, even without exogenous exposure, as a next step, we examined the impact of 4PYR on the effect of anticancer therapy with cyclophosphamide. We used tumor-bearing mice (Control) as a control, and the study groups were tumor-bearing mice treated with CP (+CP), 4PYR (+4PYR), or a combination of both (+4PYR+CP).

In this study, we also noted, similar to that observed in previous work, the effects of 4PYR accumulation—the shift of endothelial extracellular nucleotide enzymes and nicotinamide metabolism changes. Nucleotides and their metabolites found in the extracellular space play a crucial role in regulating inflammation and immune responses [21]. While extracellular nucleotides promote inflammation, their catabolites help to reduce it. Extracellular adenosine exhibits significant antithrombotic and vasodilating effects [22]. We observed a decreased rate of ATP hydrolysis but a significantly higher rate of extracellular adenosine deamination on the aortic surface of 4PYR-treated mice with breast cancer. The activity of vascular ecto-adenosine deaminase (eADA) is enhanced substantially during endothelial activation and vascular inflammation, resulting in lower adenosine bioavailability and a reduction in adenosine receptor-dependent signaling pathways [20]. Additionally, our other research showed that blocking eADA has positive effects in experimental breast cancer [23].

Furthermore, our results showed significant disturbances in nicotinamide metabolism in cancer mice treated with 4PYR. We observed a tendency towards higher NA concentration, but considerably lower NR and accumulation of one of the final products of nicotinamide metabolism—Met2PY, but not Met4PY. Nicotinamide has been indicated to have various anti-inflammatory effects [24], such as inhibiting inducible NO synthase (iNOS), scavenging free radicals, and suppressing the expression of MHC class II and intracellular adhesion molecule ICAM-1 on endothelial cells [25]. It is converted into MetNA by the cytoplasmic enzyme nicotinamide methyltransferase (NNMT). N-methylnicotinamide degrades to Met2PY and Met4PY under aldehyde oxidase (AO) [26]. We also noticed a reduction in serum concentration of NR, the other important NAD precursor, in 4PYR-treated mice compared to control breast cancer mice. Nicotinamide riboside (NR) is recognized as a significant NAD+ precursor, as it raises NAD+ levels, boosts sirtuin (SIRT) activity, enhances mitochondrial function, and strengthens the regenerative capabilities of stem cells [27]. Nicotinamide metabolism products, particularly Met2PY and Met4PY, are structurally similar to known potent PARP inhibitors such as 3-aminobenzamide [28]. Inhibition of PARP-1 activity may benefit cells because preserving the NAD pool in cells may prevent oxidative stress from damaging endothelial cells. On the contrary, such an effect may be detrimental because PARP is crucial for maintaining genome integrity [29].

In additional experiments, where we compared healthy mice to 4PYR-treated animals, we found that 4PYR has contributed to significantly increased expression of cytokines related to inflammation and cancer progression, such as CCL3, CXCL1, CCL5 or IL-1A and IL-1B. Cytokines may potentially participate in tumor initiation, elongation, and progression, as well as metastasis, angiogenesis, and therapeutic resistance development. They are also associated with increased cancer symptoms together with reduced quality of life in cancer patients [30,31].

Our study confirmed the anticancer action of CP therapy. We observed a significant reduction in the tumor volume and final tumor mass in CP-treated mice compared to Control animals. CP is an oxazaphosphorine anticancer agent used in the treatment of various hematopoietic and solid malignancies [32].

Interestingly, we observed an effect of prolonged 4PYR exposure on the cyclophosphamide anticancer action. Mice treated with both 4PYR and CP were characterized by less formation of 4PYR derivatives in organs such as the spleen, brain, or heart, a very significantly increased rate of vascular adenosine deamination, but especially by higher tumor volume and final tumor mass as compared to CP treated animals. We assumed that the observed changes could be the disruption of CP metabolism by increased 4PYR concentration in the liver. Cytochrome P450 enzymes responsible for CP metabolism are mainly CYP1A1, CYP1A2, CYP2A6, CYP2B6, CYP2C8, CYP2C9, CYP2C19, CYP2D6, CYP2E1, CYP3A4, and CYP4A11 [33,34]. We measured the levels of three of them: CYP2A6, CYP2C9, and CYP3A4. Many reports indicate that CYP3A4 and CYP2C9 are the major CYPs responsible for the metabolism of CP [35,36]. The role of CYP2A6 in this process, on the one hand, seems to be less crucial; on the other hand, the estimated contribution of CYP2A6 in CP activation is 20–40% [37,38]. We noted a variable effect of 4PYR treatment on the hepatic levels of these proteins. 4PYR exposure did not affect CYP2A6 concentrations in either the +4PYR or +4PYR+CP groups compared to +CP, but caused significantly increased CYP2C9 concentrations and reduced CYP3A4 concentrations in the +4PYR and +4PYR+CP groups as compared to +CP. Therefore, the effect of 4PYR on CP metabolism is not clear, but we believe that the most important finding of this work is that 4PYR regulates cytochrome P450 activity in general. Cytochrome P450 metabolizes about 70% of various drugs, not only anticancer ones. Cytochrome P450 enzymes are important in metabolizing steroid-based drugs, fat-soluble vitamins, fatty acids, food additives, pesticides, industrial chemicals, chemical carcinogens, and other chemicals. In general, over 90% of all oxidations and reductions of chemicals known today are catalyzed by cytochrome P450 enzymes [39]. The accumulation of 4PYR may be related to the fact that 4PYR is synthesized with AO involvement primarily in the liver. Still, it has also been shown that AO is overexpressed in malignant tumors, including breast cancer. However, Hayat F. et al. indicated that AO is not responsible for the formation of 4PYR but that it is formed as a result of excessive oxidation of NAD^+^ [40], which may be related to the fact that oxidative stress is implicated in the pathogenesis of breast cancer [41]. Considering that increased 4PYR formation occurs in the course of various pathologies, not only breast cancer, and may potentially affect the hepatic metabolism of drugs, we suggest that the concentration of this metabolite should be measured in order to select an appropriate therapeutic strategy.

## 4. Materials and Methods

### 4.1. Mice

Female BALB/c mice obtained initially from Jackson Lab (USA) were used for the experiments. Throughout the experiment, each mouse was housed in an individually ventilated cage (23 ± 1 °C, 40 ± 10% humidity) with a 12/12 h light/dark cycle with unlimited access to food and water. Experiments were conducted following a Guide for the Care and Use of Laboratory Animals by the European Parliament, Directive 2010/63/EU, and were approved by the Local Ethical Committee for Animal Experiments (46/2013).

### 4.2. Cell Culture

Cells were cultured in RPMI 1640 (IITD, Poland) with Opti-MEM^®^ (Life Technologies, Carlsbad, CA, USA) (1:1 *v*/*v*) medium with 10% fetal bovine serum (HyClone, Thermo Fisher Scientific Inc., Waltham, MA, USA), supplemented with 4.5 g/L glucose, 2 mM glutamine, 1.0 mM sodium pyruvate (all from Sigma-Aldrich, Taufkirchen, Germany) and antibiotics (penicillin and streptomycin—Polfa Tarchomin, Warszawa, Poland). Cell cultures were maintained at 37 °C in a humidified atmosphere with 5% CO_2_. Before the transplantations, cells were trypsinized, centrifuged (200× *g*, 4 °C, 5 min), suspended in fresh culture medium, and counted. Next, a cell suspension in Hanks fluid (IITD, Wroclaw, Poland) was prepared with a density of 6 × 10^6^.

### 4.3. Orthotopic Murine Breast Cancer Model

A total of 40 female BALB/c mice at 11 weeks of age were randomly divided into four groups: PBS-treated mice injected with 4T1 cancer cells (Control, *n* = 10), cyclophosphamide-treated mice injected with 4T1 cancer cells (CP, *n* = 10), 4PYR-treated mice injected with 4T1 cancer cells (4PYR, *n* = 10) and 4PYR and cyclophoshamide-treated mice injected with 4T1 cancer cells (4PYR + CP, *n* = 10). The 4T1 tumor cell suspension diluted in sterile PBS was subcutaneously injected (0.05 mL, 3 × 10^5^ cells/mouse) in the right armpit. Based on the structure analysis described earlier, 4PYR was synthesized, and the studies carried out confirmed that it is the same compound that was observed in the plasma of patients with chronic kidney disease [12]. Details of the chemical synthesis procedures are provided in the Appendix A. 4PYR (100 mg/kg/24 h) was administrated subcutaneously every 12 h after 4T1 cell injection. The injection dose of 4PYR was selected to ensure that the levels of 4PYR metabolites were lower than those found in chronic renal illness with severe malfunction and closer to those found in human diseases. Based on our earlier study that described 4PYR kinetics in rodents, which considered its incredibly quick renal excretion, the frequency of 4PYR administration was determined. Cyclophosphamide (Endoxan, Baxter Oncology GmbH, Frankfurt am Main, Germany) was administered intraperitoneally (25 mg/kg) every 48 h since day 7 of the experiment. The tumor was detected palpably after 4 days of induction. The weight of each mouse and the tumor size were measured every 2 days starting from the 4th day of tumor inoculation. The tumor was measured with a caliper, and its volume was calculated using the following formula: V (mm^3^) = (a × b^2^)/2, where a and b represent maximum and minimum diameter, respectively. Twenty-eight days after the inoculation of cancer cells, mice were weighed and anesthetized with ketamine-xylazine (100 mg/kg/10 mg/kg) by intraperitoneal injection. Blood and plasma samples, aortas, livers, muscles, spleens, hearts, brains, and lungs were collected for analysis of nucleotides, nicotinamide metabolites, and 4PYR with its derivative concentration, as well as extracellular nucleotide catabolism enzymes.

For the evaluation of the effects of 4PYR on the proinflammatory cytokine release under physiological conditions, in additional experiments, healthy control (*n*  =  5) and 4PYR-treated (*n*  =  5) mice were used. After 21 days of 4PYR (100 mg/kg/24 h) or PBS subcutaneous administration, serum samples were collected to measure the expression of cytokines associated with inflammation or cancer progression: CCL19, CCL3, SDF-1α, CX3CL1, CCL5, FAS-ligand, IL-1β and IL-1α.

### 4.4. Determination of 4PYR Metabolite Concentration in Blood and Tissues

To determine blood 4PYR metabolite concentration, the whole blood samples were immediately frozen in liquid nitrogen and extracted with 1.3 M HClO_4_ (ratio 1:1), followed by centrifugation (20,800× *g*/15 min/4 °C). Supernatants were then collected and brought to pH 6.0–6.5 using 3M K_3_PO_4_ solution. After 15 min incubation on ice, samples were centrifugated at the same conditions (20,800× *g*/15 min/4 °C), and the supernatants were analyzed using high-performance liquid chromatography (HPLC).

Livers, muscles, spleens, hearts, brains, and lungs were frozen in liquid nitrogen after collection and freeze-dried. Specimens were then extracted with 0,4 M HClO_4_ (ratio 1:25) with a glass homogenizer, centrifuged (20,800× *g*/15 min/4 °C), and the supernatants were brought to pH 6.0–6.5 with 2M KOH. After 15 min incubation on ice, samples were centrifugated under the same conditions (20,800× *g*/15 min/4 °C), and the supernatants were analyzed using high-performance liquid chromatography (HPLC; LC system, Agilent Technologies 1100 series, Santa Clara, CA, USA) on a 15 cm/4.6 mm Hypersil BDS C18 3 lm column (Thermo Fisher Scientific, 28103-023006), as previously described [42]. The distinctive UV absorption spectra (range: 210–310 nm) and retention times compared to the standards were used to identify 4PYR and its derivatives. The quantitative analysis was conducted using an external calibration using pure chemicals that were synthesized by chemical synthesis [12]. A chromatographic data system, ChemStation (Agilent Technologies, Santa Clara, CA, USA), was used to integrate and quantify the sample peaks. The Appendix A provide a detailed chromatogram of HPLC analysis (Appendix A).

### 4.5. Evaluation of Extracellular Catabolism of Adenine Nucleotides on the Aortic Surface

The aortic fragments were harvested, rinsed with 0.9% NaCl, and dissected from the surrounding tissues. Aortic sections were cut longitudinally to expose the endothelial surface and analyzed for the activities of extracellular adenine nucleotide catabolism enzymes, as previously described [43]. Aortic sections were placed in wells of the 24-well plates with 1 mL of Hanks Balanced Salt Solution (HBSS). The aortas were pre-incubated at 37 °C for 15 min. Substrates appropriate for each extracellular enzyme were sequentially added to the medium: 50 µM adenosine triphosphate (ATP) for ecto-nucleoside triphophate diphosphohydrolase (eNTPD), responsible for ATP hydrolysis, 50 µM adenosine monophosphate (AMP) for ecto-5′-nucleotidase (e5′NT), responsible for AMP hydrolysis and 50 µM adenosine for ecto-adenosine deaminase (eADA), responsible for adenosine deamination. After 0, 5, 15 and 30 min of incubation (37 °C), samples of 50 µL were collected. Following the incubation with each substrate, the medium was removed and replaced by a fresh one. When determining ATP and AMP hydrolysis rates, an adenosine deaminase inhibitor EHNA (erythro-9-(2-hydroxy-3-nonyl) adenine) was added to the buffer at a concentration of 5 µM. Before the analysis, samples were centrifuged (20,800× *g*/10 min/ 4 °C). The conversion of the substrates into the products was measured by high-performance liquid chromatography (HPLC), as previously described [42]. The reaction rates were normalized to the aorta surface area estimated using ImageJ Software (version 1.5.4). The data are shown in nmol/min/cm^2^.

### 4.6. Determination of Mice Plasma Nicotinamide Metabolites

To assess the serum nicotinamide metabolite concentration, a 50 µL serum aliquot was extracted with acetonitrile at a 1:2.4 ratio and then centrifuged (20,800× *g*/10 min/4 °C). Supernatants were collected and freeze-dried. The resulting precipitate was dissolved in water to match the original serum volume. The extraction method has been validated previously and used in many of our earlier studies [44,45]. The concentrations of nicotinamide metabolites were measured using high-performance liquid chromatography–mass spectrometry (LC/MS), as previously described [46]. The system contained a Surveyor MS pump and a degasser connected to a TSQ Vantage triple quadrupole mass detector. Heated electrospray ionization in positive mode was used. For the separation, a 50 × 2 mm Synergi Hydro-RP 100 with a particle size of 2.5 µm column was used. The mobile phase comprises water with 5 mM nonafluoropentanoic acid (A) and acetonitrile with 0.1% formic acid (B). An injection volume of 2 µL was used, with the mobile phase flowing at 0.2 mL/min. 2-Chloradenosine was used as an internal standard (Appendix A). The separation of the two derivatives, Met2PY and Met4PY, was determined by their retention times (2.29 for Met2PY and 2.58 for Met4PY) and their respective fragmentation ions (m/z 67.1, 108.17 for Met2PY and 92, 136.2 for Met4PY). A detailed chromatogram of LC/MS analysis is provided in the Appendix A.

### 4.7. Serum Cytokine Profile and Cytochrome P450 Enzyme Determination

RayBio Mouse Cytokine Antibody Array G Series 3 glass slide (RayBiotech, Norcross, GA, USA) was used for determining the relative levels of 62 factors in the serum samples, which included IL-1α, IL-1β and SDF-1α, according to the manufacturer’s instructions. CYP2A6, CYP2C9, and CYP3A4 concentrations (QY-E21827, QY-E21829, QY-E21830, Qayee-bio, Shanghai, China) were determined using enzyme-linked immunosorbent assay kits according to the manufacturer’s protocols.

### 4.8. Statistical Analysis

The results are presented as mean ± SEM. The statistical analysis used Graph Pad Prism 8 (Graph Pad, La Jolla, CA, USA). Paired and unpaired Student *t*-tests were used for comparisons between the two groups. Two-way analysis of variance with post hoc Tukey test was used to compare more than two groups. A *p*-value < 0.05 was considered a significant difference.

## 5. Conclusions

4PYR accumulation in the blood, which translates into the formation of intracellular nucleotide derivatives (4PYMP, 4PYTP, 4PYRAD), interferes with cyclophosphamide anticancer activity, most likely by affecting its metabolism by cytochrome P450 enzymes. This observation may have crucial implications for the activity of various anticancer drugs metabolized by cytochrome P450. Moreover, 4PYR induces a pro-inflammatory shift of endothelial extracellular enzymes and nicotinamide metabolism.

## Figures and Tables

**Figure 1 ijms-26-00035-f001:**
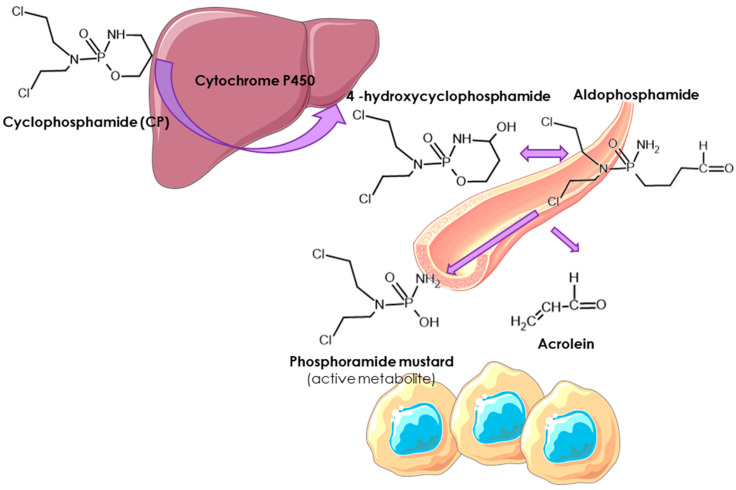
Cyclophposphamide metabolic pathway. Hepatic cytochrome P450 enzymes activate CP (prodrug) to 4-hydroxycyclophosphamide, which is in an equilibrium state with aldophosphamide. Both metabolites diffuse into cells, where aldophosphamide is converted to phosphoramide mustard and acrolein.

**Figure 2 ijms-26-00035-f002:**
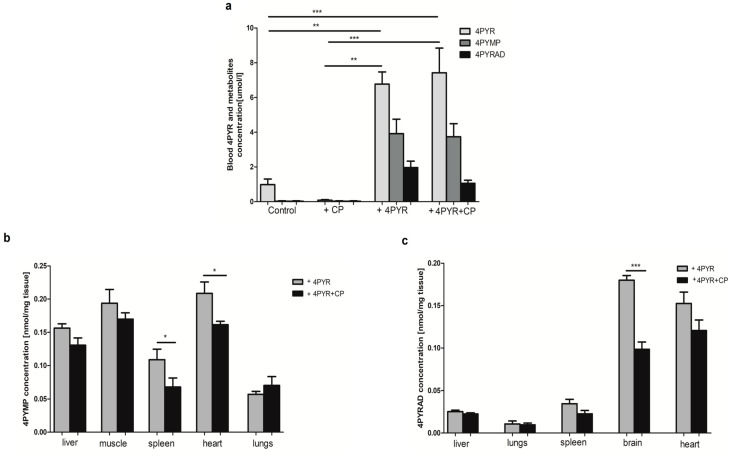
(**a**) Blood 4PYR and 4PYR metabolite concentration of mammary 4T1 carcinoma mice (Control) treated with cyclophosphamide (+CP), 4PYR (+4PYR) and 4PYR with cyclophosphamide (+4PYR+CP); (**b**) 4PYMP and (**c**) 4PYRAD concentration in the tissues of 4PYR (+4PYR) and 4PYR with cyclophosphamide (+4PYR+CP) receiving mice. Mean ± SEM, *n* = 10; two-way ANOVA with post hoc Tukey test and Student *t*-test: * *p* < 0.05; ** *p* < 0.01; *** *p* < 0.001.

**Figure 3 ijms-26-00035-f003:**
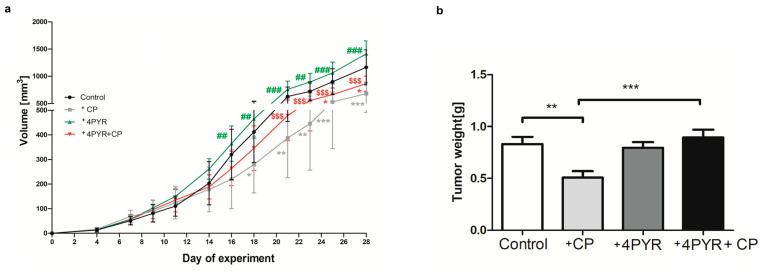
(**a**) The kinetics of murine mammary 4T1 carcinoma tumor growth and (**b**) tumor weight of mammary 4T1 carcinoma mice (Control) treated with cyclophosphamide (+CP), 4PYR (+4PYR) and 4PYR with cyclophosphamide (+4PYR+CP). Mean ± SEM, *n* = 10; two-way ANOVA with post hoc Tukey test and Student *t*-test: *** *p* < 0.001; ** *p* < 0.01; * *p* < 0.05 vs. control; ### *p* < 0.001; ## *p* < 0.01; vs. CP and 4PYR+CP and $$$ *p* < 0.001; vs. 4PYR and CP.

**Figure 4 ijms-26-00035-f004:**
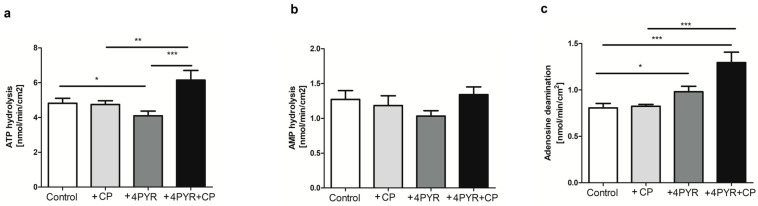
(**a**) ATP, (**b**) AMP hydrolysis and (**c**) adenosine deamination on the aorta of mammary 4T1 carcinoma mice (Control) treated with cyclophosphamide (+CP), 4PYR (+4PYR) and 4PYR with cyclophosphamide (+4PYR+CP). Mean ± SEM, *n* = 10; two-way ANOVA with post hoc Tukey test and Student t test: *** *p* < 0.001; ** *p* < 0.01; * *p* < 0.05.

**Figure 5 ijms-26-00035-f005:**
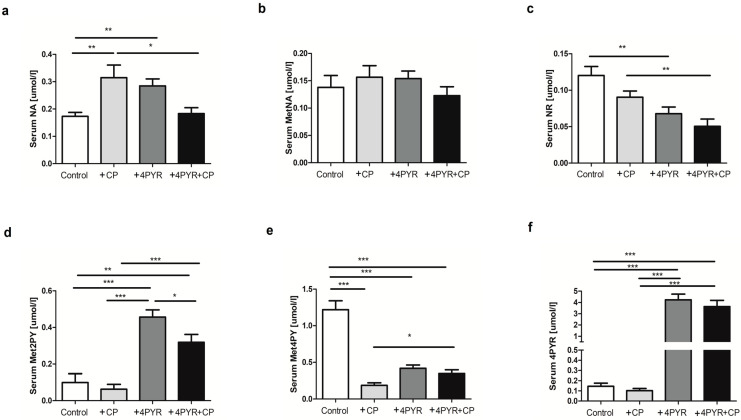
Changes in nicotinamide metabolites in the serum of mammary 4T1 carcinoma mice (Control) treated with cyclophosphamide (+CP), 4PYR (+4PYR) and 4PYR with cyclophosphamide (+4PYR+CP): the concentration of (**a**) nicotinamide (NA); (**b**) N-methylnicotinamide (MetNA); (**c**) nicotinamide riboside (NR); (**d**) N-methyl-2-pyridone-5-carboxamide (Met2PY); (**e**) N-methyl-4-pyridone-3-carboxamide (Met4PY) and (**f**) 4PYR. Mean ± SEM, *n* = 10; two-way ANOVA with post hoc Tukey test and Student *t*-test: *** *p* < 0.001; ** *p* < 0.01; * *p* < 0.05. Mean ± SEM, *n* = 10; *** *p* < 0.001; ** *p* < 0.01; * *p* < 0.05.

**Figure 6 ijms-26-00035-f006:**
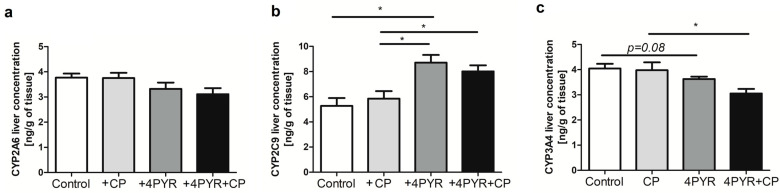
Liver concentration of cytochrome P450 enzymes involved in cyclophosphamide metabolism: (**a**) CYP2A6, (**b**) CYP2C9 and (**c**) CYP3A4 of mammary 4T1 carcinoma mice (Control) treated with cyclophosphamide (+CP), 4PYR (+4PYR) and 4PYR with cyclophosphamide (+4PYR+CP). Mean ± SEM, *n* = 5; two-way ANOVA with post hoc Tukey test and Student *t*-test: * *p* < 0.05.

**Figure 7 ijms-26-00035-f007:**
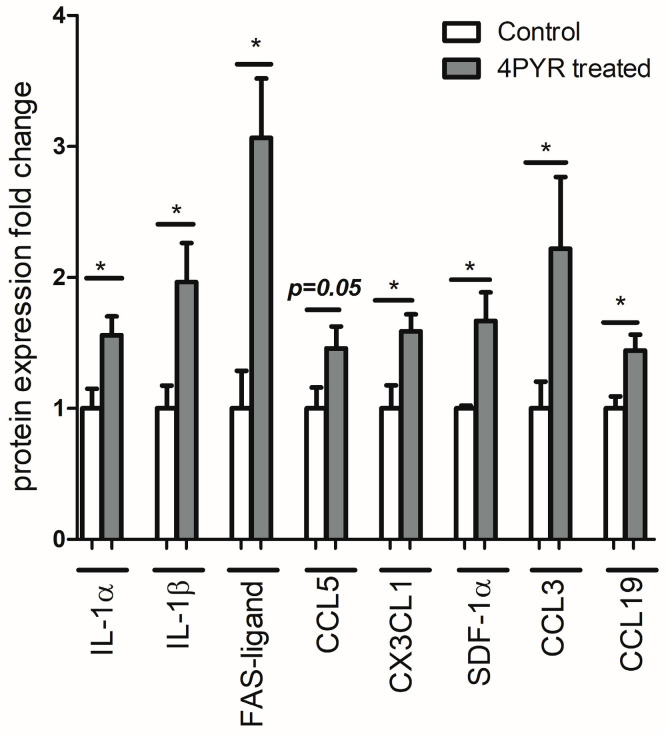
Additional measurements of the serum expression of cytokines associated with inflammation or cancer progression: CCL19, CCL3, SDF-1α, CX3CL1, CCL5, FAS-ligand, IL-1β and IL-1α in healthy mice without cancer treated with 4PYR (4PYR w/o cancer) or PBS (Control w/o cancer). Mean ± SEM, *n* = 5; Student’s *t*-test: * *p* < 0.05.

## Data Availability

The data presented in this study are available on request from the corresponding author.

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
