# Peer review of "4-Pyridone-3-carboxamide-1-β-D-ribonucleoside Reduces Cyclophosphamide Effects and Induces Endothelial Inflammation in Murine Breast Cancer Model"

_ijms, 2024, doi:10.3390/ijms26010035_

Round 1

Reviewer 1 Report

Comments and Suggestions for Authors

This study presents the first report of interference of 4PYR and its intracellular metabolites with the action of anticancer drug cyclophosphamide, resulting in increased tumor volume and mass along with increased extracellular nucleotide metabolism and vascular ecto-adenosine deaminase activity. The manuscript is well-organized and suitable for publication. However, I would like to suggest some minor changes to be included in the manuscript to make it more appealing to the readers:

1.     Kindly provide the rationale for selecting only three CYP450 isoforms (CYP2A6, CYP2C9, and CYP3A4) out of the total 12.

2.     Please note that in line 230, "aldehyde oxidase (AO)" is misspelled as "aldehide oxidase" and may need correction.

3.     It is suggested to move the general information about cyclophosphamide from lines 251-261 to the introduction, as similar content is already discussed in lines 81-84.

4.     Including schematic figures illustrating the conversion of the prodrug cyclophosphamide into its active metabolite, along with its chemical structures, could greatly enhance clarity regarding its metabolic pathway.

5.     To ensure a focused conclusion, the information about CYP enzymes in lines 282-287 might be better placed earlier in the discussion or omitted entirely.

Reviewer 2 Report

Comments and Suggestions for Authors

Comments to authors

The authors investigated the effects of “4-pyridone-3-carboxamide-1-β-D-ribonucleoside” on cyclophosphamide, noting its potential to reduce the drug's efficacy while also inducing inflammation in endothelial cells within a murine breast cancer model. Drawing from their previous research, the authors have identified that the accumulation of 4PYR in cancerous tissues may contribute to the progression of the disease. They established specific experimental setups for this study. However, a significant limitation of the work is the lack of a clearly articulated hypothesis supported by robust scientific evidence. For instance, it remains unclear necessity of a non-therapeutic molecule (referred to as 4PYR) that acts as an antagonist to an established therapeutic regimen.

Here are some constructive comments for the authors to consider:

1. What is the underlying rationale for the design of this study? Specifically, why were cyclophosphamide chosen as the focus concerning 4PYR?

2. Given that cyclophosphamide possesses significantly more reactive functional groups, have the authors investigated any potential quenching effects it may have when interacting with 4PYR?

3. The statement, “Considering the massive formation of 4PYR and its derivatives during cancer progression, it is surprising that there is still limited understanding of the metabolic and functional implications of 4PYR in this context,” may mislead readers regarding the study’s scientific premise. It should be noted that the accumulation of 4PYR and its metabolites in the bloodstream occurs only after the administration under +4PYR conditions. In addition, as evidenced by the in vivo results displayed in Figure 1a, the cancer-induced control animals did not exhibit any significant accumulation of 4PYR. This raises the question of the necessity for exogenous administration of 4-pyridone in cancer therapy.

4. In the abstract figure, it would be beneficial to include the full form of abbreviations for clarity.

5. Detailed HPLC conditions should be specified to enhance understanding of the methodology.

6. Section 4.6 requires further clarification regarding the extraction process. The authors mentioned, “To assess the serum concentration of nicotinamide metabolites, a 50 μl serum aliquot was extracted with acetonitrile in a 1:2.4 ratio and then centrifuged (20800 g for 10 min at 4 ºC). The supernatants were collected and freeze-dried.” It is essential to confirm whether freezing alone is sufficient for the removal of acetonitrile from the samples.

7. What internal standards were utilized for the LC/MS analysis?

8. It would be helpful to provide detailed chromatograms for both LC/MS and HPLC analyses.

9. On what basis was the dosage and frequency of 4PYR set at 100 mg/kg?

10. Concerning the separation of the two derivatives, Met2PY and Met4PY, which were identified by their retention times (3.22 minutes for Met2PY and 3.36 minutes for Met4PY) and corresponding fragmentation ions (m/z 108.2 for Met2PY and 136.2 for Met4PY), clarification is needed. The molecular formula for both derivatives is C7H8N2O2 (MW=152). Please explain how this molecular structure leads to the observed fragmentation of ions and provide supporting LC/MS data.

11. The synthetic methods for 4PYR are not detailed in the manuscript. The authors state, “Based on the structure analysis previously described, 4PYR was synthesized, and studies confirmed that it matches the compound found in the plasma of patients with chronic kidney disease.” It is crucial to provide either this information or cite relevant references to support this claim.
